# MODEL-AGNOSTIC WATERMARKED IMAGE RESTORATION WITHOUT ADDITIONAL TRAINING

## ABSTRACT

Post-processing image watermarking technology, which can prove the authenticity of real images, causes quality degradation and information loss in the original image. Although various methods have been proposed to restore a watermarked image to the original image, these methods are model-dependent. In this study, we propose a model-agnostic watermarked image restoration method that requires no additional training. The proposed method first extracts a message from a watermarked image and embeds the same message into the watermarked image. Then, our method computes a watermark component as the subtraction between the watermarked image and the double watermarked image. Finally, the proposed method generates a restored image by subtracting the watermark component from the watermarked image because the watermark component has a high correlation with the subtraction between the watermarked image and the original image. Experimental results show that the proposed method obtains a restored image with higher image quality for 10 of 11 existing watermarking methods. Furthermore, we have extended the existing eight methods and added a re-watermarking function that updates an embedded watermark with another watermark.

## 1 INTRODUCTION

Digital image watermarking is a technique that embeds other information into an image to protect copyright information or prove the authenticity of the image (Singh & Chadha, 2013; Wan et al., 2022; Hosny et al., 2024). In recent years, owing to advancements in image generation and editing technology using artificial intelligence (AI), content abuse and falsification through deepfakes have become increasingly sophisticated. (Luo et al., 2025). Therefore, the watermarking techniques are important in many applications to prevent the spread of false or misinformation and to provide traceability of who handled the image (Khan et al., 2014; Qasim et al., 2018). Image watermarking methods are classified into in-processing and post-processing methods (An et al., 2024), each of which has different suitable use cases. In-processing methods (Wen et al., 2023; Ci et al., 2024; 2025) integrate watermarking into an image generation model to protect AI-generated images. Furthermore, post-processing methods (Zhu et al., 2018; Zhang et al., 2019; Sander et al., 2025), which embed watermarks in an existing image, are more versatile because they are also beneficial for real photos and digital art.

Post-processing watermarking causes quality degradation in the original image and reduces data reliability (Qasim et al., 2018). Hence, a reliable watermarking technique must provide traceability and retain the information contained in the original image. According to existing research, reversible watermarking techniques (Chris W et al., 2001; Tang et al., 2024; Chen et al., 2025) can recover the embedded watermark and the original image, and TrustMark_RM (Bui et al., 2023a) can remove the watermark and restore the watermarked image embedded by TrustMark to the original image. However, these techniques are model-dependent; that is, what information is embedded and how it is embedded depend on the method. If any watermarking technique has a restoration function, it is more versatile because we can choose the appropriate embedding method according to the use case.

In this study, we propose a model-agnostic watermarked image restoration method that requires no additional training. This method leverages a common embedding policy for post-processing watermarking, where the watermark embedding process targets reducing the difference between the original image and the watermarked image. A watermark component is defined as the subtraction

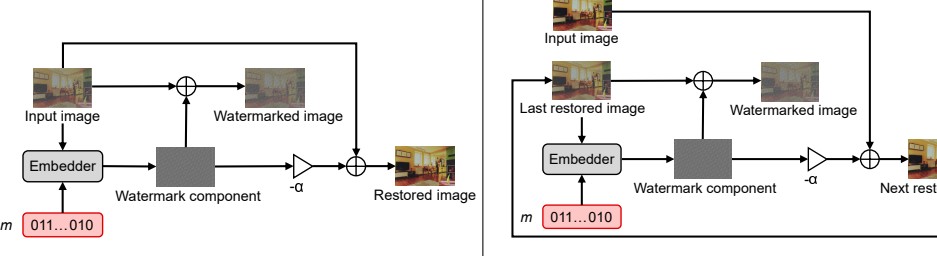

|The initial process.|The process of loop strategy.|

Figure 1: Proposed image restoration process. First, proceed as shown on the left. The already watermarked input image is watermarked again with the same method and messages. The watermark component is the subtraction between the input image and the watermarked image. The restored image is generated by subtracting the watermark component multiplied by the scaling factor $\alpha$ from the input image. Then, proceed as shown on the right. The last restored image is used for computing the watermark component, and the next restored image is generated similarly. Each time process on the right is repeated, the restored image becomes increasingly similar to the original image.

between an input image and an output image in the watermark embedding process and a first watermark component is defined as the subtraction between an original image and a watermark image. Furthermore, this method obtains a double watermarked image by re-embedding the same message into the watermarked image and computes a second watermarking component as the subtraction between the watermarked image and the double watermarked image. The watermark component is assumed to be dependent on an input image when a watermarking method and a message to be embedded are fixed. In this situation, the first watermark component and the second watermark component are highly similar because the watermarked image is highly similar to the original image according to the embedding policy. As a result, we compute the second watermark component using the watermarked image and restore the watermarked image to the original image by subtracting the second watermark component from the watermarked image. We conduct experiments on watermarked image restoration by 11 existing post-processed methods, including state-of-the-art studies. Experimental results indicate that the proposed method improves restored image quality in 10 out of 11 watermarking methods. In particular, we obtained improvements of 6 to 20 dB in PSNR for the three latest watermarking such as WAM (Sander et al., 2025), MaskMark (Hu et al., 2025a), and CRMark (Chen et al., 2025). Moreover, experimental results indicate that the proposed method adds a re-watermarking function similar to TrustMark(Bui et al., 2023a) with eight methods. In summary, the main contributions of this study are as follows:

- This study propose a novel watermarked image restoration method, which improves both watermark removal accuracy and image quality, without additional training.

- Experiments with practical state-of-the-art watermarking methods show that the proposed method enhances the versatility of existing post-processing watermarking methods by making them re-watermarkable.

- This study reveal new knowledge that the common embedding policy for post-processing watermarking is useful for watermarked image restoration.

## 2 RELATED WORKS

### 2.1 WATERMARK EMBEDDING

Imperceptibility of image watermarking technology has been generally discussed (Zhong et al., 2023). Existing post-processing watermarking approaches commonly reduce the difference between the watermarked image and the original image to achieve imperceptibility. Traditional watermarking methods embed information into spatial or frequency domains (Al-Haj, 2007; Cox et al., 2007). Deep learning has considerably improved the robustness to image distortion and the watermarked image quality. HiDDeN (Zhu et al., 2018) is the first end-to-end trained deep network that uses an

encoder-decoder architecture. A noise layer, which distorts encoded images inserted between the encoder and the decoder, improves the robustness of these methods against image distortion. MBRS (Jia et al., 2021) also adopts a layer that is randomly chosen among a real JPEG, a simulated JPEG, and a noise-free layer to be robust against strong JPEG compression with a quality factor of 50. StegaTamp (Tancik et al., 2020) embeds hyperlink bit strings into photographs on the Internet and uses a spatial transformer in the decoder network to develop robustness against small perspective changes. RivaGAN (Zhang et al., 2019) integrates an attention module with two independent adversarial networks that critique video quality and optimize robustness for video processing operations such as scaling, cropping, and compression. TrustMark (Bui et al., 2023a) is a generative adversarial network (GAN) based method with spatial and spectral losses to balance the trade-off between watermarked image quality and watermark recovery accuracy. SSL (Fernandez et al., 2022) inserts watermarks into the latent space of images using pre-trained self-supervised networks without specific training for watermarking. RoSteALS (Bui et al., 2023b) embeds a message into the latent space of a frozen pretrained autoencoder, where a noise model is inserted between the image decoder and the secret decoder to improve the robustness against image modification. RAW (Xian et al., 2024) introduces learnable watermarks directly into both of frequency and spatial domains in the original image data and trains an encoder and a binary classifier for watermark presence simultaneously.

The aforementioned methods embed a message into an entire image or a defined area of an image. Furthermore, WAM (Sander et al., 2025) and MaskMark (Hu et al., 2025a) provide local watermarking, which can embed any region of an image and extract the watermarked area. WAM performs data augmentation with a mask of an arbitrary area between the encoder and the decoder in training, and the decoder outputs a mask indicating a watermarked area simultaneously with a message. Moreover, it can detect multiple messages within the same image. MaskMark inputs an image, a message, and a concatenated mask region, and includes a distortion pool between the encoder and decoder to make it robust against image manipulation. In addition, WAM and MaskMark use a Just-Noticeable-Difference (JND) module, which modulates the embedding signal based on human visual sensitivity to enhance the image perceptual quality.

### 2.2 IMAGE RESTORATION

Image restoration techniques have been extensively studied in image processing and computer vision (Elad et al., 2023; Jiang et al., 2025). The existing techniques have addressed correcting a specific type of image degradation such as denoising, dehazing, and deblurring, but they have not targeted watermarked image restoration. To address this issue, model-dependent restoration techniques have been proposed to achieve restoration for watermarked images embedded in a specific method. Watermarked image restoration techniques are important to prevent the loss of image information due to degradation by watermarking, particularly in the medical, legal, and military fields (Khan et al., 2014). For example in the medical field, to prevent errors in diagnosis and treatment caused by the modification of the medical images, reversible watermarking techniques, which simultaneously restore the embedded watermark and the original image, have been developed (Qasim et al., 2018). Integer-to-integer wavelet transform (Lee et al., 2007) has been proposed to ignore rounded error. Integer discrete flows (Hoogeboom et al., 2019) has used a bijective integer map as auxiliary information to ignore rounded errors. CRMark (Chen et al., 2024; 2025) introduces an Integer invertible watermark network by leveraging integer discrete flows, and has achieved high recovery accuracy and speed-up by embedding auxiliary information reversibly into an image. TrustMark (Bui et al., 2023a) provides a watermark removal model TrustMark_RM for re-watermarking, where two models are trained for watermark embedding and watermark removal. The removal model generates a restored image from a watermarked image using KBNeT (Zhang et al., 2023) of the image restoration model, which removes the embedded watermark as noise. Furthermore, a watermarked image is updated by embedding another watermark in the restored image, and the latest embedded watermark can be extracted after 20 updates.

### 2.3 WATERMARK REMOVAL

Watermark removal techniques aim at generating an image from which no watermark is extracted from a watermarked image. An et al. (2024) have classified watermark removal techniques into three approaches: distortion attacks, regeneration attacks, and adversarial attacks. Distortion attacks include geometric distortion such as rotation and cropping, photometric distortion such as brightness,

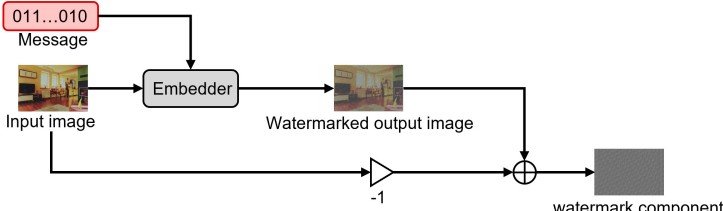

Figure 2: Illustration of the watermark component. In post-processing watermarking, the embedder outputs a watermarked image from an input image and a message. The watermark component is defined as the subtraction between the input image and the output image.

contrast, and color adjustments, and degradation distortions such as Gaussian blur and compression. State-of-the-art watermarking methods are robust to photometric distortion but not geometric distortion, particularly rotation (Sander et al., 2025). In addition, Zhao et al. (2024) have confirmed that image compression techniques (Ballé et al., 2018; Cheng et al., 2020) reduce watermark detection accuracy. Regeneration attacks generate an un-watermarked image from a watermarked image. Saberi et al. (2024) have proposed a diffusion purification attack, which uses a diffusion model to add noise to a watermarked image and then reconstruct a watermark removed image. Zhao et al. (2024) have also shown the effectiveness of a diffusion model for watermark removal. First, a distortion attack breaks the watermark; subsequently, a diffusion model regenerates an image similar to the original image using the broken image as initial noise. Adversarial attacks include two approaches: the embedding attack that perturbs latent representations, and the surrogate detector attack that trains proxy models to deceive watermark detectors (An et al., 2024). Both approaches have shown that Tree-Ring (Wen et al., 2023) of an in-processing method is vulnerable to adversarial attacks. The watermark detection process for in-processing methods depends on the detector to decode and verify messages from the watermarked image and can be disturbed by perturbations on image embedding. An attacker acquires many watermarked and un-watermarked images to train a surrogate detector and forward attacks against it to the actual watermarked detector. In another approach, Watermark Steganalysis (Yang et al., 2024) provides steganalysis attacks by using subtraction between the averages of the original and watermarked images to remove the watermark. The authors have defined a graybox attack that allows access to both the original and watermarked images and a blackbox attack that does not allow access to the original images. As a result, they revealed that content-agnostic watermarking approaches are vulnerable to this attack, but content-adaptive ones are not.

## 3 PROPOSED METHOD

As mentioned in the preceding section, post-processing watermarking methods aim to reduce the difference between the original image and the watermarked image. We leverage this embedding policy to propose the restoration process shown as Figure 1.

In WAM (Sander et al., 2025) and MaskMark (Hu et al., 2025a), the watermarked output image $I_{wm}$ with the message $m$ embedded in the original image $I_{orig}$ is represented as follows:

$$I_{wm} = I_{orig} + \mu \times \text{JND}(I_{orig}) \odot \delta_\theta(I_{orig}, m), \tag{1}$$

where $\delta_\theta$ is the watermark signal of the model output, and $\mu$ is the JND scaling factor to control the watermark strength. This watermark embedding process is generalized as follows:

$$I_{wm} = I_{orig} + W(I_{orig}, m), \tag{2}$$

where $W$ is a watermark component that depends on the $I_{orig}$ and $m$ and illustrated in Figure2, In our process, we firstly consider embedding the same message into the watermarked image, then define the double watermarked image as $I_{dwm}$, as shown in the following equation:

$$I_{dwm} = I_{wm} + W(I_{wm}, m). \tag{3}$$

Because the message is fixed, the watermark component depends on the input image. The difference between $I_{orig}$ and $I_{wm}$ is sufficiently small because the difference is small by the embedding pol-

Table 1: Experimental results of the image restoration. The column of Embedded (Emb.) indicates the results with the watermarked images for comparison. In each method, PSNR and bit accuracy is calculated by averaging 5000 pairs of original and restored images.

| Method | PSNR ($\uparrow$) | | | | Bit accuracy ($\sim 0.5$) | | | |
|---|---|---|---|---|---|---|---|---|
| | Emb. | DiffAtt | WmStg | Ours | Emb. | DiffAtt | WmStg | Ours |
| DwtDct | 37.80 | 24.51 | 36.77 | 41.27 | 0.895 | 0.500 | 0.899 | 0.919 |
| DwtDctSvd | 37.87 | 24.44 | 36.87 | 40.11 | 0.999 | 0.645 | 0.973 | 0.715 |
| HiDDeN | 36.73 | 20.16 | 35.75 | 40.91 | 0.999 | 0.502 | 0.969 | 0.963 |
| RivaGAN | 40.53 | 24.50 | 37.75 | 42.66 | 0.998 | 0.620 | 0.748 | 0.816 |
| MBRS | 42.46 | 22.02 | 40.30 | 54.01 | 0.997 | 0.513 | 0.981 | 0.552 |
| SSL | 39.90 | 21.99 | 38.36 | 39.90 | 0.999 | 0.600 | 0.996 | 0.978 |
| RoSteALS | 27.25 | 21.93 | 26.12 | 27.82 | 0.994 | 0.769 | 0.289 | 0.766 |
| TrustMark | 42.05 | 24.47 | 41.83 | 46.25 | 1.000 | 0.662 | 0.818 | 0.532 |
| WAM | 38.68 | 24.45 | 37.96 | 45.34 | 1.000 | 0.660 | 0.970 | 0.584 |
| MaskMark | 41.50 | 24.57 | 41.39 | 61.05 | 1.000 | 0.625 | 0.741 | 0.430 |
| CRMark | 40.24 | 21.99 | 40.25 | 53.45 | 1.000 | 0.702 | 0.892 | 0.521 |
| WAM (local) | 44.67 | 24.52 | 43.63 | 51.38 | 0.999 | 0.663 | 0.969 | 0.597 |
| MaskMark (local) | 44.36 | 24.53 | 43.45 | 57.28 | 0.999 | 0.523 | 0.991 | 0.499 |

icy. Therefore, the first watermark component $W(I_{orig}, m)$ and the second watermark component $W(I_{wm}, m)$ have a high correlation as follows:

$$W(I_{orig}, m) \approx \alpha \times W(I_{wm}, m), \tag{4}$$

where $\alpha$ is a scaling factor, which is evaluated in Section 4 and discussed in Section 5. Then, the restored image $I_{res}$ is defined as follows:

$$\begin{aligned} I_{res} &= I_{wm} - \alpha \times W(I_{wm}, m) \\ &= I_{orig} - (\alpha \times W(I_{wm}, m) - W(I_{orig}, m)), \end{aligned} \tag{5}$$

where the bottom row derived from (2) shows $I_{res}$ approximately equal to $I_{orig}$ due to (4). In addition, we do not need to compute the watermark component because (5) is transformed into the following:

$$I_{res} = I_{wm} - \alpha \times (I_{dwm} - I_{wm}). \tag{6}$$

The process can be looped, as shown in Figure 1 on the right, by making the restored image the next image to be watermarked. Because the last restored image is more similar to the original image than the input image, the next restored image has higher quality than the previous one. Hence, when the number of loops of this process increases, the quality of the recovered image increases. The optimum number of loops is evaluated in Section 4 and discussed in Section 5. In addition, the proposed method realizes re-watermarking like TrustMark (Bui et al., 2023a) by embedding another watermark in the restored image.

## 4 EXPERIMENTS

### 4.1 EXPERIMENTAL SETUP

We evaluate the proposed method using the COCO2017 (Lin et al., 2014) validation dataset including 5000 images on 10 post-processing watermarking methods: DwtDct (Al-Haj, 2007), DwtDctSvd (Cox et al., 2007), HiDDeN (Zhu et al., 2018), RivaGAN (Zhang et al., 2019), MBRS (Jia et al., 2021), SSL (Fernandez et al., 2022), RoSteALS (Bui et al., 2023b), TrustMark (Bui et al., 2023a), WAM (Sander et al., 2025), MaskMark (Hu et al., 2025a), and CRMark (Chen et al., 2025). Because the input image size and the bit strings length to be embedded are different for each method, input images are resized to $256 \times 256$ for HiDDeN, MBRS, SSL, RoSteALS, and CRMark, and $512 \times$



| Original | 1st WC | 2nd WC | 1.7× 2nd WC | 3rd WC |

Figure 3: Visualisation of the restoration process in the case of TrustMark. The first watermark component (1st WC) is $10\times$ the subtraction between the watermarked image and the original image, the second watermark component (2nd WC) is $10\times$ the subtraction between the double watermarked image and the watermarked image, and the third watermark component (3rd WC) is the subtraction between the restored image and the original image. The optimum scale factor is 1.7.



| 1st loop | 2nd loop | 3rd loop | 4th loop | 5th loop |

Figure 4: Visualization of the loop strategy in the case of TrustMark. Each image indicates $10\times$ the subtraction between the restored image for each loop and the original image.

512 for the others. In the cases of DwtDct and DwtDctSvd, the restoration process is applied in wavelet domain. Additionally, for WAM and MaskMark, we evaluate local watermarking, which embeds a watermark in a rectangular region of 25% area of the original image with a random location and aspect ratio. Our experiments were conducted on an Intel Core Ultra 9 285K Processor and an Nvidia RTX 4090 GPU.

The evaluation of restoration is based on the image quality difference between the original image and the restored image used for the following image metrics: peak signal to noise ratio (PSNR), structural similarity index measure (SSIM), learned perceptual image patch similarity (LPIPS) (Zhang et al., 2018), and single image frechet inception distance (SIFID) (Rott Shaham et al., 2019). We report PSNR in the main text and SSIM, LPIPS, and SIFID in the Appendix. In addition, bit accuracy is also computed to evaluate to assess whether the embedded watermark remains in the image.

We compare the performance of the proposed method with two model-agnostic watermark removal technologies: the diffusion model based attacking (DiffAtt) (Zhao et al., 2024) and Watermark Steganalysis (WmStg) (Yang et al., 2024). In the setup of Watermark Steganalysis, we evaluate a greybox pattern, in which the $N$ pairs of original and watermarked images are first averaged, then subtracted to extract the watermark pattern, and finally the watermark pattern is subtracted from the watermarked image for removal. $N$ is the number of images to be averaged, and we set $N = 5$ in our experiments because Zhao et al. (2024) describes that a smaller $N$ increases the effectiveness of this method. We also compare the restoration performance with TrustMark (Bui et al., 2023a) and CRMark (Chen et al., 2025). TrustMark performed 20 times embedding and restoration iterations following the experimental procedures described by Bui et al. (2023a).

## 4.2 WATERMARKED IMAGE RESTORATION

We first determine the optimum value of the scaling factor $\alpha$ to maximize the PSNR between the original image and the watermarked image for each method. Figure 5 shows that there is an optimum value to make the restored image closer to the original image. We obtain the optimum $\alpha$ that maximizes the PSNR for each method as follows: 1.0 for DwtDct, 1.0 for DwtDctSvd, 0.8 for HiDDeN, 1.0 for RivaGAN, 1.2 for MBRS, 0.3 for SSL, 0.5 for RoSteALS, 1.7 for TrustMark, 0.5 for WAM, 1.1 for MaskMark, and 1.0 for CRMark. Figure 3 shows the effect of the scaling factor. The second watermark component is affected by the watermark already embedded. For example of

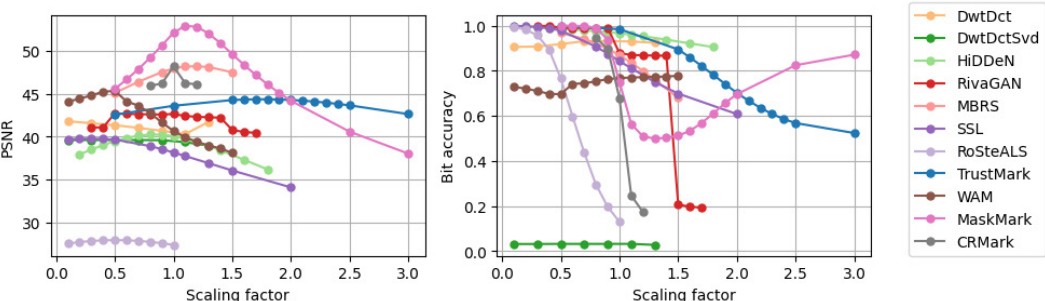

Figure 5: PSNR and bit accuracy when the scale factor varies. The scaling factor that gives the highest PSNR is selected as the optimal value.

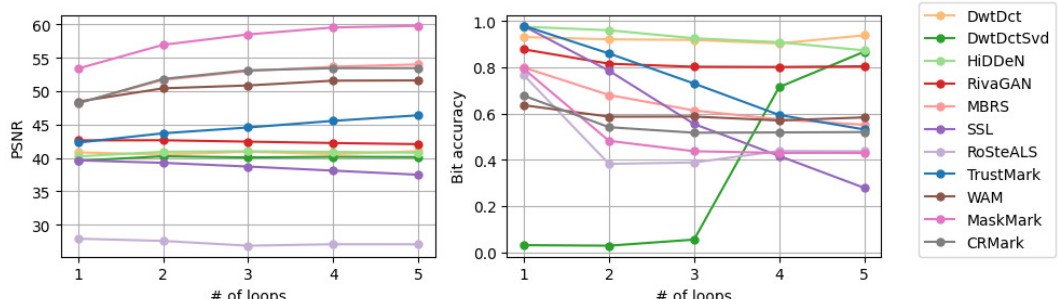

Figure 6: Evaluation results of the loop strategy. Each figure shows PSNR or bit accuracy when the number of loops increases.

TrustMark, the second watermark component is smaller than the first, and therefore the optimum scaling factor is larger than 1.

Then, we assess the optimum number of loops of the loop strategy. Figure 6 shows the evaluation results when the maximum number of loops is 5. TrustMark, WAM, MaskMark, and CRMark denote that the PSNR increases when the number of loops increases, and thus indicate that loop processing contributes to the improvement of the watermark image restoration. In other methods, the PSNR decreased as the number of loops increased. The optimum number of loops to maximize the PSNR is as follows: 1 for RoSteALS and SSL, 2 for RivaGAN, 3 for DwtDct and HiDDeN, 4 for DwtDctSvd, and 5 for the others. Figure 4 shows the effect of the loop. The difference decreases as the number of loops increases. For example of TrustMark, the PSNR increases to 46.92 on the fifth loop from 45.10 on the first loop.

Finally, Table 1 shows restoration performances compared to model-agnostic attackers. The proposed method has a higher PSNR compared to that of Emb., that is, restoration of the watermarked image to the original is successful. Furthermore, the diffusion model based attacking and Watermark Steganalysis have degradation of image quality. The diffusion model based attacking considerably loses image quality to achieve watermark removal, while Watermark Steganalysis is not successful in image quality improvement. In terms of bit accuracy, watermark removal is successful in five methods, including MBRS, TrustMark, WAM, MaskMark, and CRMark. Table 2 shows the result of the comparison with TrustMark_RM and CRMark. Our method is superior to TrustMark_RM but inferior to CRMark. CRMark can output a restored image equal to the original image, and thus its PSNR is infinity. In addition, CRMark embedding failed 4.9% of the time, even though no other methods failed to embed.

### 4.3 RE-WATERMARKING

In the re-watermarking evaluation, the process of embedding a different random message each time followed by image restoration is repeated 20 times. The evaluation points are how accurately the last embedded message is extracted and how much the image quality degrades from the original

Table 2: Restoration performance compared with model-dependent methods.

| Method | PSNR ($\uparrow$) | Bit accuracy ($\sim 0.5$) | Process time [ms] |
|---|---|---|---|
| TrustMark_RM | 41.90 | 0.715 | 545.45 |
| TrustMark with the proposed method | 46.25 | 0.532 | 244.25 |
| CRMark | Inf. | 0.499 | 1170.40 |
| CRMark with the proposed method | 53.45 | 0.521 | 2183.65 |

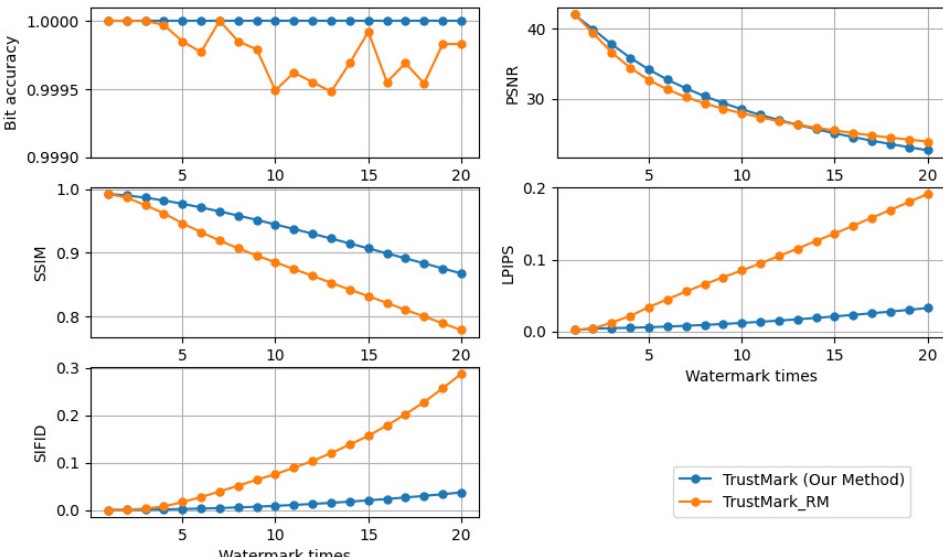

Figure 7: Re-watermarking performance of the proposed method compared to TrustMark_RM. Re-watermarking is performed 20 times for each method. Each figure shows bit accuracy or image quality metric for the $N$-th($N \leq 20$) embedding.

image. Figure 7 shows that TrustMark_RM has a slight degradation of bit accuracy but the proposed method maintains. In SSIM, LPIPS, and SIFID, the proposed method achieves significantly better results than TrustMark_RM. Table 3 shows PSNR and bit accuracy of the 20th watermarked image. We report SSIM, LPIPS, and SIFID in the appendix. The proposed method maintains bit accuracy of eight methods except for SSL, RoSteALS, and WAM. PSNR varies by watermarking method, but none is significantly worse than TrustMark_RM. Therefore, the proposed method successfully adds a re-watermarking function equivalent to TrustMark_RM to eight methods.

## 5 DISCUSSION

**Scale factor and loop strategy.** As described in Section 3, we have introduced the scaling factor and the loop strategy to improve restoration performance. The scaling factor has the effect of approximating the second watermark component embedded in the watermarked image to the first one embedded in the original image. If the watermark does not affect the watermarked image, the optimal scale factor is equal to 1; however, experimental results have given different optimal values depending on the method. Therefore, the scale factor might indicate how much the embedding process is affected by noise in the image if the embedded watermark is considered noise to the original image. Furthermore, the loop strategy has an effect on reducing the difference between the restored image and the original image. Our assumption is that the restored image quality improves when the number of loops increased; however, the experimental results show that only five methods such as MBRS, TrustMark, WAM, MaskMark, and CRMark are applicable. These five methods are newer watermarking methods. With the development of watermarking technology, the newer methods successfully achieve the embedding policy regarding the smaller difference between the original image

Table 3: Performance of the 20th re-watermarking. If the method is in bold, our method is used.

| Method | PSNR (↑) | Bit acc. (↑) | Method | PSNR (↑) | Bit acc. (↑) |
|---|---|---|---|---|---|
| TrustMark_RM | 23.84 | 0.9998 | **TrustMark** | 22.59 | 1.0000 |
| **DwtDct** | 34.79 | 0.9991 | **DwtDctSvd** | 35.05 | 0.9998 |
| **HiDDeN** | 25.89 | 0.9999 | **RivaGAN** | 22.45 | 0.9997 |
| **MBRS** | 40.00 | 0.9993 | **SSL** | 26.22 | 0.7016 |
| **RoSteALS** | 27.25 | 0.7987 | **CRMark** | 38.12 | 1.0000 |
| **WAM** | 34.73 | 0.9335 | **WAM (local)** | 36.60 | 0.9234 |
| **MaskMark** | 40.52 | 1.0000 | **MaskMark (local)** | 40.77 | 0.9993 |

and the watermarked image. Hence, the fact that the proposed method using this embedding policy fits these methods well indicates that the assumption of the authors in this paper is effective in watermarking technology

**Comparison with reversible watermarking.** Table 2 shows CRMark is highly accurate in terms of restoration. However, CRMark has two disadvantages: it fails to embed a watermark in some images and takes a long time to process. Because our method can be applied to various methods, it is possible to select the most suitable watermarking method in consideration of bit length to be embedded, input image size, process time, image quality, and so forth. The proposed approach does not outperform CRMark, but extends the use case for watermarking.

**Restoration process as an attacker.** The proposed restoration methods have the effect of watermark removal against some watermarking methods; therefore, we consider the possibility of using it as an attack method. There are two types of watermark attacking methods: white-box setting (Jiang et al., 2023; Hu et al., 2025b) in which the attacker has access to the watermarking model, and black-box setting (Müller et al., 2025) in which the attacker cannot have access to the model. The proposed method corresponds to white-box setting, additionally, an attacker needs to accurately reproduce the setup when the watermark is embedded when using the proposed method as a watermark attack method. Therefore, it is considered that only the person who originally embedded the watermark can perform watermark removal. In summary, the proposed method is effective in enhancing the usefulness of watermarking, not in attacking the watermark.

**Limitations.** The watermarked image restoration method can be applied to only post-processing watermarking, and not to in-processing watermarking. Although the proposed method can obtain restored images for various post-processing watermarks, it does not completely restore the image quality degradation caused by the watermark. Therefore, we considered that there is a coexistence relationship with a reversible watermark, which is model-dependent and restores completely the original image.

## 6 CONCLUSION

In this study, we proposed a novel model-agnostic method for watermarked image restoration without additional training. The proposed approach leverages the re-embedding of the same watermark message to estimate and subtract the watermark component to restore a watermarked image to an original image. The experiments on the COCO2017 dataset and 11 different post-processing watermarking techniques have shown that the proposed method improves image quality for 10 methods and achieves re-watermarking for eight methods. Results show that the proposed method adds image restoration to existing post-processing watermarking methods and further improves the versatility of watermarking technology. We believe that this study will contribute to a deeper understanding of watermark embedding and lead to further progress in watermarking research.

## REPRODUCIBILITY STATEMENT

Section 4 provides the specifics of our evaluation setup. The proposed method is implemented by making minor modifications to the GitHub repository published by the authors of existing watermarking methods. The URL of each GitHub repository is listed in the reference section. We provide a supplementary material, which is a Python code to be added to the GitHub repository of WAM (Sander et al., 2025), as an example of implementation of the proposed method. No other changes to the WAM repository are required as of the submission date.

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

## A    ADDITIONAL EXPERIMENTAL RESULTS

### A.1    PROCESS TIME

Table 4 shows process time for a image for each watermarking methods. Our experiments were conducted on an Inter Core Ultra 9 285K Processor and an Nvidia RTX 4090 GPU. Our restoration method depends on embedding time.

Table 4: Process time per a image.

| Method | Process Time [ms] Embedding | Decoding | Method | Process Time [ms] Embedding | Decoding |
|---|---|---|---|---|---|
| DwtDct | 16.63 | 9.56 | DwtDctSvd | 47.36 | 29.44 |
| HiDDeN | 3.16 | 2.74 | RivaGAN | 13.01 | 12.95 |
| MBRS | 4.57 | 3.80 | SSL | 288.98 | 5.15 |
| RoSteALS | 4.46 | 3.98 | TrustMark | 48.85 | 12.68 |
| WAM | 10.95 | 15.88 | MaskMark | 5.30 | 15.35 |
| CRMark | 2183.65 | 1170.40 | | | |

### A.2    IMAGE RESTORATION

Experimental results of the image restoration with SSIM, LPIPS, and SIFID. In all metrics, our method approximates the quality of the watermarked image to that of the original image.

Table 5: SSIM of the restored image.

| Method | SSIM ($\uparrow$) Emb. | DiffAtt | WmStg | Ours |
|---|---|---|---|---|
| DwtDct | 0.966 | 0.692 | 0.951 | 0.978 |
| DwtDctSvd | 0.975 | 0.693 | 0.962 | 0.985 |
| HiDDeN | 0.975 | 0.555 | 0.961 | 0.981 |
| RivaGAN | 0.978 | 0.691 | 0.974 | 0.985 |
| MBRS | 0.989 | 0.623 | 0.977 | 0.998 |
| SSL | 0.980 | 0.615 | 0.967 | 0.978 |
| RoSteALS | 0.834 | 0.620 | 0.725 | 0.834 |
| TrustMark | 0.996 | 0.696 | 0.996 | 0.995 |
| WAM | 0.979 | 0.687 | 0.975 | 0.991 |
| MaskMark | 0.995 | 0.695 | 0.993 | 0.998 |
| CRMark | 0.988 | 0.692 | 0.987 | 1.000 |
| | | | | |
| WAM (local) | 0.995 | 0.695 | 0.993 | 1.000 |
| MaskMark (local) | 0.985 | 0.617 | 0.986 | 0.998 |

Table 6: LPIPS of the restored image.

| Method | LPIPS ($\downarrow$) | | | |
| --- | --- | --- | --- | --- |
| | Emb. | DiffAtt | WmStg | Ours |
| DwtDct | 0.01087 | 0.13283 | 0.01423 | 0.00899 |
| DwtDctSvd | 0.01724 | 0.13819 | 0.02211 | 0.01835 |
| HiDDeN | 0.00738 | 0.17864 | 0.01304 | 0.00404 |
| RivaGAN | 0.03634 | 0.14817 | 0.04393 | 0.02313 |
| MBRS | 0.00399 | 0.16376 | 0.01161 | 0.00094 |
| SSL | 0.02498 | 0.17677 | 0.03281 | 0.02375 |
| RoSteALS | 0.04447 | 0.17593 | 0.15452 | 0.03810 |
| TrustMark | 0.00141 | 0.12861 | 0.00177 | 0.00273 |
| WAM | 0.03600 | 0.15112 | 0.04502 | 0.00995 |
| MaskMark | 0.00948 | 0.13338 | 0.01212 | 0.00248 |
| CRMark | 0.02651 | 0.14145 | 0.02414 | 0.00008 |
| WAM (local) | 0.01154 | 0.13357 | 0.01387 | 0.00030 |
| MaskMark (local) | 0.01620 | 0.17089 | 0.01675 | 0.00036 |

Table 7: SIFID of the restored image.

| Method | SIFID ($\downarrow$) | | | |
| --- | --- | --- | --- | --- |
| | Emb. | DiffAtt | WmStg | Ours |
| DwtDct | 0.01434 | 0.09225 | 0.01871 | 0.00876 |
| DwtDctSvd | 0.01003 | 0.09489 | 0.01494 | 0.01357 |
| HiDDeN | 0.02286 | 0.36915 | 0.02407 | 0.00420 |
| RivaGAN | 0.07235 | 0.11712 | 0.09364 | 0.02663 |
| MBRS | 0.00121 | 0.14097 | 0.00215 | 0.00005 |
| SSL | 0.02571 | 0.15843 | 0.03476 | 0.02797 |
| RoSteALS | 0.03734 | 0.18125 | 0.08786 | 0.02935 |
| TrustMark | 0.00135 | 0.08644 | 0.00149 | 0.00534 |
| WAM | 0.08199 | 0.13170 | 0.09674 | 0.01153 |
| MaskMark | 0.01148 | 0.09293 | 0.01503 | 0.00151 |
| CRMark | 0.03369 | 0.11702 | 0.02596 | 0.00001 |
| WAM (local) | 0.00931 | 0.09521 | 0.01158 | 0.00004 |
| MaskMark (local) | 0.02306 | 0.15869 | 0.02057 | 0.00013 |

## A.3 RE-WATERMARKING

Figure 8 shows re-watermarking performance of the proposed method for existing 11 watermarking methods. Table 8 shows the performance of the 20th watermarked image including SSIM, LPIPS, and SIFID. We have obtained similar results to PSNR, in SSIM, LPIPS, and SIFID. RoSteALS performed worse than the other methods due to the high degradation of image quality from a single embedding. The characteristics differ depending on the method, however, our restoration method can be applied to various watermarks, therefore we can select the most suitable method.

Figure 9 shows examples of watermark components for each watermarking method. The original image is the same as in Figure 3. In each watermarking method, the third watermark component (3rd WC), which indicates the subtraction between the restored image and the original image, is more inconspicuous than the first watermark component (1st WC), which indicates the subtraction between the watermarked image and the original image.

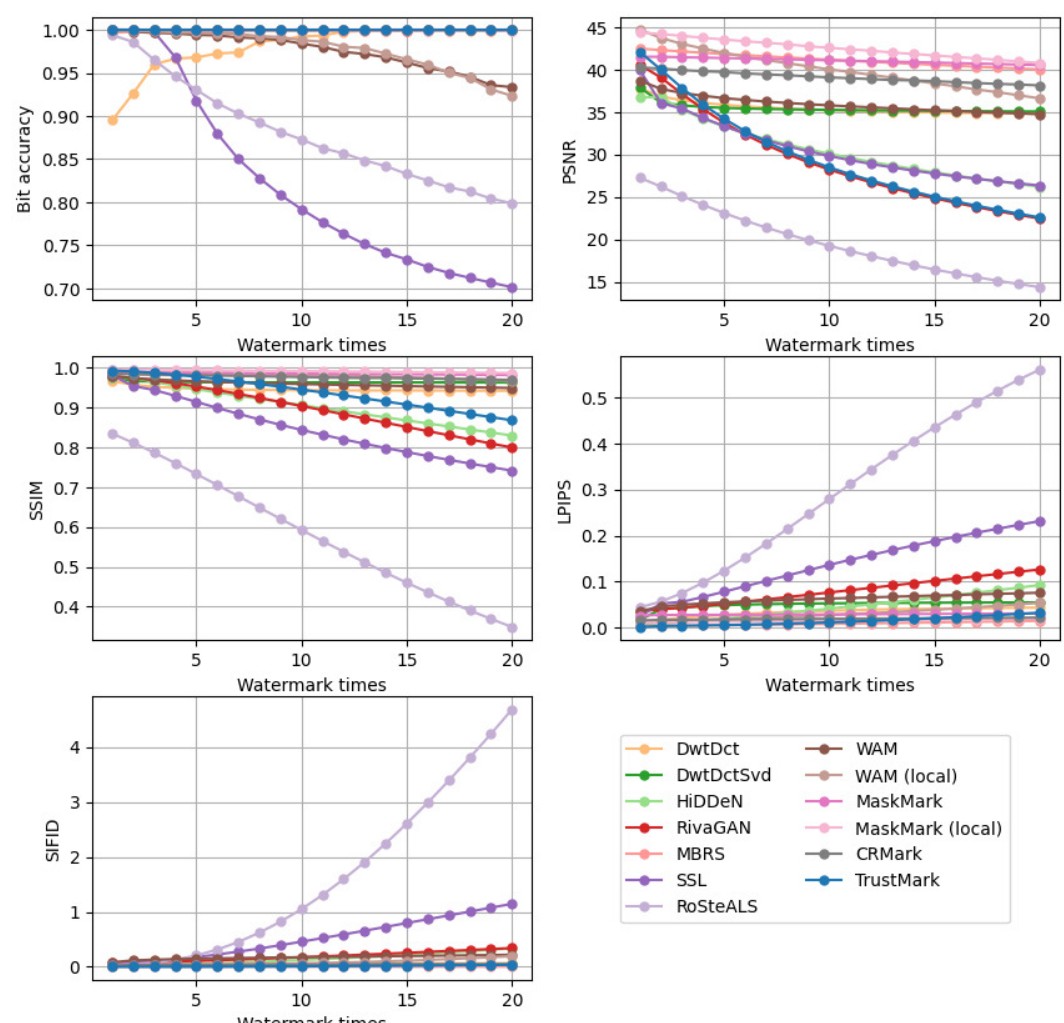

Figure 8: Re-watermarking performance with the proposed methods. Re-watermarking is performed 20 times for each method.

Table 8: Performance of the 20th re-watermarking. If the method is in bold, our method is used.

| Method | PSNR (↑) | SSIM (↑) | LPIPS (↓) | SIFID (↓) | Bit accuracy (↑) |
|---|---|---|---|---|---|
| TrustMark_RM | 23.84 | 0.779 | 0.19158 | 0.28739 | 0.9998 |
| **TrustMark** | 22.59 | 0.868 | 0.03254 | 0.03708 | 1.0000 |
| **DwtDct** | 34.79 | 0.942 | 0.04380 | 0.06536 | 0.9991 |
| **DwtDctSvd** | 35.05 | 0.963 | 0.05511 | 0.04528 | 0.9998 |
| **HiDDeN** | 25.89 | 0.822 | 0.09828 | 0.35974 | 0.9999 |
| **RivaGAN** | 22.45 | 0.799 | 0.12645 | 0.34050 | 0.9997 |
| **MBRS** | 40.00 | 0.969 | 0.01517 | 0.00306 | 0.9993 |
| **SSL** | 26.35 | 0.737 | 0.23550 | 1.18042 | 0.7016 |
| **RoSteALS** | 27.25 | 0.349 | 0.55981 | 4.68643 | 0.7987 |
| **CRMark** | 38.12 | 0.969 | 0.02209 | 0.04114 | 1.0000 |
| **WAM** | 34.73 | 0.949 | 0.07605 | 0.21813 | 0.9335 |
| **WAM (local)** | 36.60 | 0.967 | 0.05419 | 0.19363 | 0.9234 |
| **MaskMark** | 40.52 | 0.981 | 0.03120 | 0.04236 | 1.0000 |
| **MaskMark (local)** | 40.77 | 0.986 | 0.02016 | 0.02805 | 0.9993 |

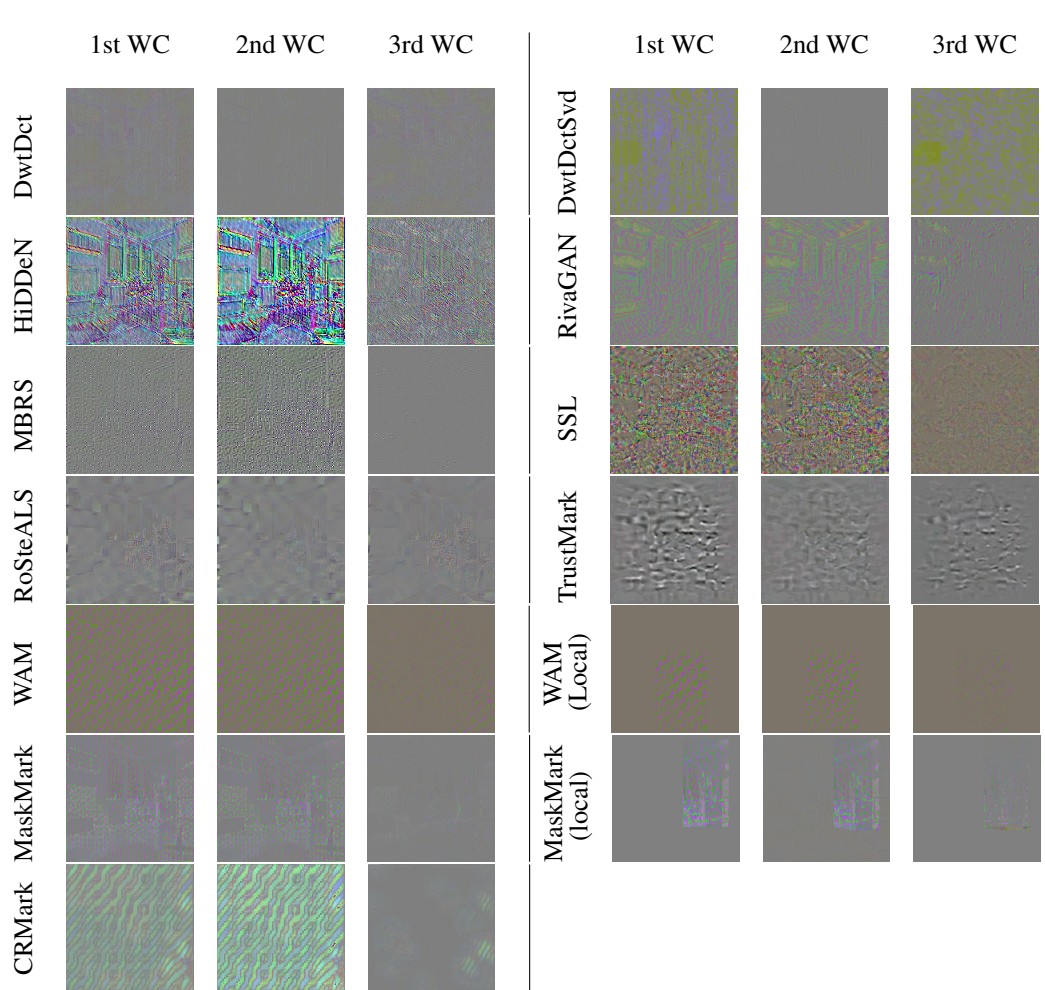

Figure 9: Visualisation of watermark components. 1st WC, 2nd WC, and 3rd WC represent the same as in Figure 3.

