# OpenReview forum: "Model-agnostic Watermarked Image Restoration without Additional Training"
_ICLR.cc/2026/Conference — Submitted to ICLR 2026_

### Official Review · Reviewer_87CM · 2025-10-28

**Soundness:** 2
**Presentation:** 2
**Contribution:** 2
**Rating:** 4
**Confidence:** 2

**Summary:**

The paper proposes a model-agnostic watermarked image restoration method, which aims to address the issue that post-processing image watermarking technology causes quality degradation and information loss in the original image. Unlike existing model-dependent restoration techniques, such as reversible watermarking techniques and TrustMark RM (Bui et al., 2023a), a core advantage of this method is that it requires no additional training.

**Strengths:**

- **Clear writing:**  The paper is well-written and has a logical flow.
- **Clear Baseline Comparison:**  The paper evaluates 11 post-processed watermarking methods. The baseline comparison covers major strategies in watermark removal.

**Weaknesses:**

- **Discrepancy between White-Box Setting and Practical Usefulness:**  Although the paper labels the method as "model-agnostic", this primarily means it requires no additional training. Fundamentally, the method operates under a strict white-box setting: The user must possess the same embedder and  message m. This requirement contradicts the threat models of many generic watermark removal attacks, such as DiffAtt (Zhao et al., 2024) and Watermark Steganalysis (Yang et al., 2024), which often deal with black-box scenarios where the embedder/message is inaccessible. Furthermore, the proposed restoration method can only be applied to post-processing watermarking, and not to in-processing watermarking. Please clarify these scope boundaries more explicitly.
- **Fairness of Baseline Comparison (White-Box vs. Black-Box/Grey-Box):**  The comparison pits a white-box recovery method against generic attack methods designed for black-box or grey-box settings, specifically DiffAtt (Zhao et al., 2024) and WmStg (Yang et al., 2024). White-box methods inherently hold a significant information advantage, leading to expectedly higher image quality metrics. Please analyze this discrepancy more carefully.
- **Weak Theoretical Foundation for Watermark Component Correlation:**  The core approximation supporting the method is the high correlation assumption:  $W(I_{\text{orig}}, m) \approx \alpha \cdot W(I_{\text{wm}}, m)$. However, the watermark component W often results from complex or non-linear operations, such as deep learning models or complex frequency transformations. While the paper presents visualizations demonstrating the components' similarity, it lacks formal mathematical proof or boundary analysis to guarantee the robustness of this crucial approximate linear relationship across diverse and non-linear watermark embedders. Such theoretical analysis would strengthen this work.

**Questions:**

- **Inconsistencies in Reference Formatting:**  The review notes that the list of references contains a mix of different formats and types of entries, such as conference papers, arXiv preprints, and embedded GitHub URLs. Please standardize to the conference’s bibliography style.
- Other issues as noted above under Weaknesses.

---

> ### Author Response · Authors · 2025-11-21
>
> Thank you very much for your valuable feedback. We will incorporate the theoretical analysis of the points you raised and work to clarify the effectiveness of the proposed methodology.
>
> >Discrepancy between White-Box Setting and Practical Usefulness:
>
> This approach focused primarily on watermark updates, resulting in insufficient insights regarding its potential as an attack method. We will incorporate analysis of its use as an attack method and conduct quantitative comparisons.
>
> >Fairness of Baseline Comparison (White-Box vs. Black-Box/Grey-Box):
>
> To effectively evaluate the proposed methodology as an attack technique, we will compare it with white-box methods and conduct a multifaceted assessment.
>
> >Weak Theoretical Foundation for Watermark Component Correlation:
>
> We have recognized that this study is limited to experimental investigation and lacks sufficient theoretical analysis. We will conduct experiments and analysis on assumptions other than high correlation to strengthen the theoretical foundation of the proposed method.
>
> >Inconsistencies in Reference Formatting:
>
> We apologize for the lack of attention to the reference format. We will take great care with the citation style going forward.

---

### Official Review · Reviewer_AhPt · 2025-10-30

**Soundness:** 2
**Presentation:** 2
**Contribution:** 2
**Rating:** 2
**Confidence:** 4

**Summary:**

This paper proposes a model-agnostic watermarked image restoration method that requires no additional training to address the quality degradation caused by post-processing image watermarking. The proposed approach first extracts a message from a watermarked image and re-embeds the same message into it, then computes a watermark component through subtraction between the original watermarked image and the double-watermarked version. By leveraging the high correlation between this component and the difference between the watermarked and original image, the method restores the image by subtracting the watermark component.

**Strengths:**

1. It is a  training-free method. Watermark restoration can be achieved without additional training, reducing computational costs and time consumption.
2. Compared to other model-agnostic watermark removal technologies, it demonstrates significantly superior performance.

**Weaknesses:**

1. This method is technically limited to recovering images by extracting watermark residuals, demonstrating weak innovation.
2. The introduction and related work sections of this paper are  difficult to read, containing numerous logical inconsistencies that make it hard for readers to grasp the author's intended arguments.
3. What does "information loss" mean in the abstract, and how does "Post-processing watermarking causes quality degradation in the original image and reduces data reliability" from the introduction relate to this? Specifically, what does data reliability refer to.

**Questions:**

1. This method must ensure that the embedded information remains identical each time. However, in practice, variable time information is often included, resulting in differing embedded data each time. How can this be resolved?
2. This method assumes that the acquired watermark image is free from distortion interference. However, I am curious about the restoration effectiveness of this method when only distorted watermark images are available.
3. Section 4.2 states that different methods require varying numbers of iterations. Why does PSNR increase for some methods as iterations progress, while it decreases for others? Does this indicate limitations in the method?
4. Watermarks like SSL and RoSteALS embedded in latent space appear to yield very low PSNR upon recovery. Does this method perform poorly on such watermarking techniques? Additionally, how does it fare on LaWa[1]?

[1] Rezaei A, Akbari M, Alvar S R, et al. Lawa: Using latent space for in-generation image watermarking[C]//European Conference on Computer Vision. Cham: Springer Nature Switzerland, 2024: 118-136.

---

> ### Author Response · Authors · 2025-11-21
>
> Thank you very much for providing important insights. We have realized that our paper lacked sufficient logical justification. We will improve our study based on the insights you have provided.
>
> >What does "information loss" mean in the abstract, and how does "Post-processing watermarking causes quality degradation in the original image and reduces data reliability" from the introduction relate to this? Specifically, what does data reliability refer to.
>
> Information loss here refers to situations where correct information cannot be extracted from the original image due to the impact of image quality degradation caused by watermark embedding. A concrete example is when areas indicating abnormalities in medical images become unclear due to quality degradation, preventing accurate diagnosis?this represents a decline in data reliability caused by information loss. Since reduced data reliability in medical images can lead to medical errors, it is essential to prevent this.
>
> >This method must ensure that the embedded information remains identical each time. However, in practice, variable time information is often included, resulting in differing embedded data each time. How can this be resolved?
>
> This method assumes that all embedding information is known when removing or updating watermarks. Therefore, in cases where the embedded information changes based on time information, as you pointed out, the time information itself is also required. To expand the applicability of the proposed method, we will continue exploring ways to achieve watermark updates using less information.
>
> >This method assumes that the acquired watermark image is free from distortion interference. However, I am curious about the restoration effectiveness of this method when only distorted watermark images are available.
>
> We confirmed that the removal effect is maintained even for images that underwent JPEG compression or contrast adjustment after watermark embedding. However, quantitative evaluation across various models remains future works. Therefore, we will investigate quantitative assessments such as differences between watermark models and the relationship between distortion magnitude and recovery rate.
>
> >Section 4.2 states that different methods require varying numbers of iterations. Why does PSNR increase for some methods as iterations progress, while it decreases for others? Does this indicate limitations in the method?
>
> If a single operation per method fails to sufficiently remove the watermark, we believe repetition may be effective, however it is true that logical justification is lacking. To resolve this, we will add further experiments and analysis.
>
> >Watermarks like SSL and RoSteALS embedded in latent space appear to yield very low PSNR upon recovery. Does this method perform poorly on such watermarking techniques? Additionally, how does it fare on LaWa[1]?
>
> We will clarify the applicability of the proposed method by adding experiments and analysis on how its response changes depending on where the watermark is embedded within the space.

---

### Official Review · Reviewer_Djxe · 2025-10-31

**Soundness:** 2
**Presentation:** 1
**Contribution:** 1
**Rating:** 2
**Confidence:** 3

**Summary:**

This paper considers the problem of removing watermarks from watermarked images, and then restoring them.

Consider a watermarking method $W$ that takes an image $I$ and a message $m$. This paper considers watermarks that compute the watermarked image as $I_wm = I_orig + W(I_orig, m)$.  The paper proposes repeating this process to produce $I_wm' = I_orig + W(I_wm, m)$ multiple times.

They also propose restoring the watermark via creating the restored image $I_res = I_wm - alpha W(I_wm, m)$ for various choices of alpha. They evaluate their method against 11 watermarking methods from 2007, 2007, 2018, 2019, 2021, 2022, 2023, 2023, 2025, 2025, and 2025.

**Strengths:**

They compare their method to many watermarking methods.

**Weaknesses:**

1. The "method" they propose is ridiculously simple: Just apply the watermark... again! And, if you want to remove it, subtract the watermark multiple times! I'm not sure this should be a published idea, much less at a major ML conference like ICLR.

2. For lacking any meaningful insights (not to mention at techniques or theory), they try to make up for it with the experiments. But only three of the 11 watermarking methods they compare to are from 2024 or later. Given the pace of generative AI, I would recommend comparing to more methods (they even cite TreeRing, but I don't *think* they compare to it): TreeRing, RingID, Gaussian Shading, PRC, WIND, and SEAL.

3. The presentation is poor. The writing is difficult to follow (even though their idea is quite simple), and the Figure 1 and Figure 2 are a) not very informative and b) quite visually unappealing.

TreeRing: https://arxiv.org/abs/2305.20030
RingID: https://arxiv.org/abs/2404.14055
Gaussian Shading: https://arxiv.org/abs/2404.04956
WIND: https://arxiv.org/html/2412.04653v3
PRC: https://arxiv.org/pdf/2410.07369?
SEAL: https://arxiv.org/abs/2503.12172

**Questions:**

Could you please justify the watermark removal process in Equation 5?

**Details Of Ethics Concerns:**

Their method is designed to remove watermarks from images. The obvious application is bad actors using a big tech company to generate an image of something fake or harmful, removing the watermark so the image can't be traced, and then passing it off as a real image. See this article from today's New York Times about generated images used for death threats: https://www.nytimes.com/2025/10/31/business/media/artificial-intelligence-death-threats.html?unlocked_article_code=1.xk8.OrVm.yHdWPOeRxZiq&smid=nytcore-ios-share&referringSource=articleShare

I don't see an impact statement or discussion of the harmful effects of their work (though, to be fair, I didn't read the appendix).

---

> ### Author Response · Authors · 2025-11-21
>
> We really appreciate your valuable feedbacks. We have understood your points regarding the areas where the paper is insufficient, and I will strive to improve them in the future.
>
> >For lacking any meaningful insights (not to mention at techniques or theory), they try to make up for it with the experiments. But only three of the 11 watermarking methods they compare to are from 2024 or later. Given the pace of generative AI, I would recommend comparing to more methods (they even cite TreeRing, but I don't think they compare to it): TreeRing, RingID, Gaussian Shading, PRC, WIND, and SEAL.
>
> The proposed method is applicable for only post-processing watermarking and cannot be applied to the in-processing methods listed. However, this fact is merely a limitation of our study, and we will also pursue insights into in-processing in the future.
>
> >The presentation is poor. The writing is difficult to follow (even though their idea is quite simple), and the Figure 1 and Figure 2 are a) not very informative and b) quite visually unappealing.
>
> Fig. 1 and 2 merely illustrated the experimental procedure and contained little information, therefore we will create figures that clearly convey the concept of the proposed method. We will also strive to improve the presentation of the text.
>
> >Could you please justify the watermark removal process in Equation 5?
>
> We apologize for not having a clear answer. We have recognized that our work on watermark removal formulas was limited to experimental investigation and lacked sufficient theoretical analysis. We will conduct additional experiments and analysis under various watermark model assumptions to strengthen the theoretical foundation of the proposed method.
>
>
> > Details Of Ethics Concerns:
>
> Exactly as you stated, it is indeed necessary to consider it as a means of attack. While this paper was written from the perspective of watermark updating, we will also explore watermark removal to enhance the contribution of our study.

---

### Official Review · Reviewer_S96p · 2025-11-01

**Soundness:** 3
**Presentation:** 2
**Contribution:** 2
**Rating:** 2
**Confidence:** 5

**Summary:**

In this paper, the authors propose a model-agnostic watermarking restoration method by simply calculating the difference between the watermarked image and the double-watermarked image and the substract it from the watermarked image. The author also conduct a discussion about how the proposed method develop re-watermarking and how to act as an attack method. Abundant experimental result show promising results.

**Strengths:**

1. This paper contains abundant experiments on so many watermarking methods, which imporves the persuasiveness of the paper.
2. The discussion part is quite interesting. Restoration process acts as an attaker could be useful and more practical in real-world.

**Weaknesses:**

1. The proposed scheme seems to be too simple. Only simply calculating the difference between the watermarked image and the double-watermarked image and the substract it from the watermarked image. Well, we have to say, this really make sense, but we think more theoretical analysis on this shall be provided, or in other word, WHY the proposed method works. For now, we suggest the persuasiveness is not enough. Since there are too many post-processing watermarking methods, a more comprehensive analysis help the reader to believe the proposed method is truly model-agnostic.

2. No offense to this work and any other work in this area, but in our humble opinion, the watermarking restoration methods which aims to fight against quality degradation and information loss might fall into a wrong path for real-world application. Since most of the post-processing watermarking methods are used for protection or generally speaking, they act as a unique stamp to prove something (belong to whom or whether it is generated by AI etc.). Therefore, we might raise two point: first, keeping the watermarking exist is vital for most of the time (unless you are the attacker who want to bypass the watermark and use the image illegally), and second, quality degradation and information loss might not be a vital problem (unless there are severe degradation, but in recent years, most watermarking methods achieves satisfying visual quality). Therefore, we are appreciating more on the part where the author tries to become an attacker who remove the watermark and add a new one. We think that is more relevant to real-world applications.

3. Figure 1 and 2 are not good, occupying much space but offers too little information.

**Questions:**

1. It is clear that the proposed method cannot be applied to watermarking methods that is not post-processing. However, in-processing methods become more popular in recent years. Try to discuss how the proposed method can be applied on them with only a minor manipualation. (We will considered changing our score based on the response, especially on this question)

---

> ### Author Response · Authors · 2025-11-21
>
> We really appreciate your many insightful comments. We have understood that the proposed method lacks sufficient experimentation and discussion to demonstrate its effectiveness. We will strive to improve them in the future.
>
>
> >The proposed scheme seems to be too simple. Only simply calculating the difference between the watermarked image and the double-watermarked image and the substract it from the watermarked image. Well, we have to say, this really make sense, but we think more theoretical analysis on this shall be provided, or in other word, WHY the proposed method works. For now, we suggest the persuasiveness is not enough. Since there are too many post-processing watermarking methods, a more comprehensive analysis help the reader to believe the proposed method is truly model-agnostic.
>
> We will quantitatively evaluate the relationship between the differences in the characteristics of post-processing watermark models and the results of the proposed method, and conduct an analysis to logically demonstrate the effectiveness of the proposed method.
>
>
> >No offense to this work and any other work in this area, but in our humble opinion, the watermarking restoration methods which aims to fight against quality degradation and information loss might fall into a wrong path for real-world application. Since most of the post-processing watermarking methods are used for protection or generally speaking, they act as a unique stamp to prove something (belong to whom or whether it is generated by AI etc.). Therefore, we might raise two point: first, keeping the watermarking exist is vital for most of the time (unless you are the attacker who want to bypass the watermark and use the image illegally), and second, quality degradation and information loss might not be a vital problem (unless there are severe degradation, but in recent years, most watermarking methods achieves satisfying visual quality). Therefore, we are appreciating more on the part where the author tries to become an attacker who remove the watermark and add a new one. We think that is more relevant to real-world applications.
>
> The exsinting study (TrustMark) has proposed re-watermarking technology as a method to enhance content traceability by combining metadata and watermarks to manage content editing histories. This study also focuses on watermark updates. However, exactly as you stated, the proposed method could also be considered a form of attack, and our discussion from an attacker's perspective is lacking. We will do further experiments and analysis to demonstrate the effectiveness of the proposed method, considering real-world attack methods.
>
>
> >Figure 1 and 2 are not good, occupying much space but offers too little information.
>
> We have understood Fig. 1 and 2 merely illustrated the experimental procedure and contained little information. We will make figures that incorporate the theoretical background, building upon the experiments and analyses described above.
>
> >It is clear that the proposed method cannot be applied to watermarking methods that is not post-processing. However, in-processing methods become more popular in recent years. Try to discuss how the proposed method can be applied on them with only a minor manipualation. (We will considered changing our score based on the response, especially on this question)
>
> Currently, when based on the proposed method, different models are required for in-processing watermark removal during both embedding and removal. Therefore, we consider that the "minimal intervention" criterion is not satisfied. We will continue to explore this topic and strive to make this study more effective.

---

> > ### Comment · Reviewer_S96p · 2025-11-26
> >
> > Hope the raised questions could help authors better polish the paper for future venues.

---

### Meta-Review · Area_Chair_UxA5 · 2026-01-07

**Summary:**

The paper proposes a (very) simple approach for restoring watermarks in images, compares this approach against several existing watermarking methods, and shows that the methods outperforms several existing methods.

The reviewers raised several concerns about this approach. The method (simply adding the watermark again multiple times) was considered somewhat simple/trivial. The paper contained no rigorous analysis of the method, and the fundamental assumption (of high correlation) may not apply to more sophisticated watermarking schemes. Despite the numerous listed comparisons, the paper failed to compare with the latest/state of the art methods. Finally, the writing could have been much clearer. I concur with these concerns; in its current form the paper isn't suitable for publication at ICLR.

**Reviewer Concerns:**

The list of reviewer concerns included:
- the significance/impact of the method (it being too simple to be publishable)
- the lack of analysis
- the lack of comparisons to the latest state of the art in image watermarking
- the clarity of the presentation

The authors did respond to some of these concerns, but I fear that the responses were somewhat shallow and didn't substantially engage with the criticisms.

**Reviewer Scores:**

Reviewer S96p: wouldn't have changed

Reviewer Djxe: wouldn't have changed

Reviewer AhPt: wouldn't have changed

Reviewer 87CM: wouldn't have changed

---

### Decision · Program_Chairs · 2026-01-26

Reject